# Persistence of Low Back Pain and Predictive Ability of Pain Intensity and Disability in Daily Life among Nursery School Workers in Japan: A Five-Year Panel Study

**DOI:** 10.3390/healthcare12020128

**Published:** 2024-01-05

**Authors:** Megumi Aoshima, Xuliang Shi, Tadayuki Iida, Shuichi Hiruta, Yuichiro Ono, Atsuhiko Ota

**Affiliations:** 1Department of Public Health, Fujita University School of Medicine, Toyoake 470-1192, Japan; 51021001@fujita-hu.ac.jp (M.A.); 81023017@fujita-hu.ac.jp (X.S.); yono@fujita-hu.ac.jp (Y.O.); 2Department of Physical Therapy, Faculty of Health and Welfare, Prefectural University of Hiroshima, Mihara 723-0053, Japan; iida@pu-hiroshima.ac.jp; 3Research Center of Health, Physical Fitness, and Sports, Nagoya University, Nagoya 464-8601, Japan; hiruta@htc.nagoya-u.ac.jp

**Keywords:** disability in daily life, Japan, low back pain, nursery school workers, pain intensity, prevalence, panel study

## Abstract

Nursery school workers are known for having a high prevalence of low back pain (LBP). The natural history of LBP and the determinants of persistent LBP remain unclear. We examined the prevalence of persistent LBP and whether pain intensity and disability in daily life due to LBP affected the persistence of LBP among these workers. A five-year panel study was conducted for 446 nursery school workers in Japan. LBP, pain intensity, and disability in daily life due to LBP were assessed with a self-administered questionnaire survey. Pain intensity was assessed using the numerical rating scale (NRS). The Roland–Morris Disability Questionnaire (RDQ) was used to assess disability in daily life due to LBP. At baseline, 270 nursery school workers (60.5%) suffered from LBP. The estimated prevalence of persistent LBP was 84.6% (80.3–88.9%), 82.2% (77.7–86.8%), and 82.0% (77.4–86.5%) at 1, 3, and 5 years after the initial study, respectively. NRS scores of 5 or greater predicted the persistence of LBP at 1 and 3 years after the initial survey (adjusted odds ratios: 4.01 (1.27–12.6) and 8.51 (1.87–38.7), respectively), while RDQ scores did not. In conclusion, LBP highly persisted for a long time and pain intensity predicted persistent LBP among nursery school workers in Japan.

## 1. Introduction

Low back pain (LBP) is a common health condition. In 2020, it affected 619 million people worldwide [1]. The global age-standardized rate of years lived with disability (YLDs) due to LBP was 832 per 100,000 [1]. Occupational ergonomic factors are the leading factor for YLDs, accounting for them by as much as 22% [1]. Workers with LBP are more likely to take absences due to sickness and require return-to-work management [2]. Systematic reviews showed that LBP is associated with increased healthcare costs not only in high-income countries but also in low- and middle-income countries [3,4].

The importance of early childhood education and care is widely acknowledged. Early childhood education and care contribute to children’s cognitive and emotional development, learning, and well-being [5]. Simultaneously, researchers have long warned about the health and safety risks that nursery school workers are exposed to in their work environment and the health concerns resulting from the risks, i.e., infectious diseases, musculoskeletal injuries, accidents, and occupational stress [6]. Nursery school workers are exposed to the following harmful ergonomic factors while working: heavy lifting, pushing and applying force, frequent bending and twisting, awkward standing and posture, sudden load bearing, and repetitive work [6]. Since these are the supposed risk factors of LBP, LBP has attracted attention as a prevalent health concern among nursery school workers.

Cross-sectional studies have shown a high prevalence of LBP in nursery school workers [7,8,9,10,11,12,13,14]. In Japan, Tsuboi et al. [7] and Kudo and Sasaki [8] reported point prevalence rates of 43.0% (n = 142) and 71.9% (n = 57; valid response rate: 63.2%; female: 100%), respectively. Yamamoto-Kataoka et al. reported one-year and lifetime prevalence rates of 41.8% and 83.7%, respectively (n = 154; valid response rate: 38%; female: 93.5%) [9]. Isono et al. revealed that the one-month and lifetime prevalence rates were 62.0% and 85.5%, respectively (n = 333; valid response rate: 56.9%; female: 100%) [10]. In an Italian study, 70.6% of nursery school workers with musculoskeletal disorders experienced pain in their lower back (n = 677; valid response rate: 76.6%; female: 100%) [11]. In Jordan, Alghwiri et al. found that the prevalence of LBP was 46% in women and 36% in men, with 55% of women and 49% of men reporting that LBP interfered with their work (n = 439; valid response rate: 88%; female: 77%) [12]. Evidence is scarce on the natural course of LBP among nursery school workers since few longitudinal studies have been conducted for them to date. In general, pain in chronic LBP decreased by half in the first 6 weeks; after that, the decrease was gradual and pain persisted [15]. LBP persisted one year after onset in 65% of patients with non-specific LBP who were cared for by primary healthcare physicians [16]. These findings were not confirmed for nursery school workers.

Little is also known about the determinants of prolonged LBP in nursery school workers. Pain intensity and disability in daily life due to LBP are thought to affect the prognosis of LBP [17,18,19]. A recent systematic review found that female sex, high pain intensity, high body weight, carrying heavy loads at work, and difficult working positions were potential risk factors for chronicity of LBP [19]. However, there is little evidence to confirm whether this is true for nursery school workers.

We conducted a panel study to describe the natural course of LBP of nursery school workers. We also examined whether pain intensity and disability in daily life due to LBP predicted persistent LBP among them. Our purpose was to elucidate how much LBP persists among nursery school workers and whether nursery school workers with severe pain intensity and disability in daily life due to LBP suffer from persistent LBP. Policymakers and healthcare workers could utilize these findings to create strategies and implement preventative measures for LBP.

## 2. Materials and Methods

### 2.1. Study Design

This was a five-year panel study. The baseline survey was conducted in June 2015. The participants were followed up at 1, 3, and 5 years. All data were collected using a self-administered questionnaire survey. The details of the study design were partly reported elsewhere [20].

### 2.2. Subjects

Convenience sampling was performed since we had no way to access a population of nursery school workers large enough to conduct random sampling. We asked an association consisting of 36 private nursery school facilities in Nagoya, Japan, and its suburbs to recruit the participants. The inclusion criterion was being employed in nursery schools when the baseline survey was conducted. The exclusion criterion was leaving work for health-related or any other reasons when the baseline and follow-up surveys were conducted. The managers of the nursery schools introduced the survey to the workers orally and by written notice. The workers who were willing to participate in the survey gave us their consent. A bubble sheet that also contained the questions was sent to the participants by a printing company. They filled it in and sent it back to the printing company for data entry. Although we did not sum up the exact number of workers in the 36 nursery schools, approximately 20 regular workers and 20 casual workers were working on average in each nursery school. Therefore, the total number of eligible participants was estimated to be 1440. To reduce costs, we did not distribute the questionnaire to all workers but only to those who were willing to participate in this study. Of the 36 nursery schools, 34 had at least one respondent. A total of 446 nursery school workers gave their consent and participated in the baseline survey, which resulted in an estimated response rate of approximately 30%. We followed them up at 1, 3, and 5 years.

### 2.3. Study Variables

#### 2.3.1. LBP

##### Definition of LBP

We determined the participants’ statuses of LBP through self-reporting. They answered a question that asked them to specify where they were feeling LBP. This question was followed by five sub-questions: (1) feeling pain in the lower back only, (2) feeling pain in the lower and upper back, (3) feeling pain that extends to the buttock and thighs, (4) feeling pain and numbness that extends to legs and feet, and (5) feeling pain in the shoulders, neck, and/or arms as well as the lower back. The participants answered with either a “yes” or a “no” to each sub-question. Those who answered “yes” to any sub-question were classified as having LBP. Those who answered “yes” to sub-questions (3) and/or (4) were considered to have LBP with leg pain.

##### Pain Intensity

Pain intensity was assessed using the numerical rating scale (NRS) score [21,22]. Participants were asked to choose where the intensity of their current LBP fell on a scale of 0 to 10. An integer value of 0 was defined as no pain, whereas 10 was the worst pain imaginable. An NRS score of 5 or greater was defined as clinically significant pain [23].

##### Disability in Daily Life Due to LBP

We used the Japanese version of the Roland–Morris Disability Questionnaire (RDQ) [24] to evaluate the limitations in daily activities caused by LBP, such as difficulty standing, walking, sitting, dressing, and working. The questionnaire comprises 24 yes/no questions. The sum of the “yes” answers provides the RDQ score. A greater score indicates a greater disability in daily life due to LBP. If the RDQ score is 4 or greater, it is considered as having a serious disability in daily life due to LBP [25].

##### Impact of LBP on Work

To assess how LBP affected work in the last month, we provided six options: (1) “I cannot work without taking occasional days off”; (2) ”I cannot work without taking a break sometimes”; (3) ”It hurts a lot, but I do not need to take a break”; (4) ”I feel slight pain occasionally”; (5) ”I would like a break or day off, but I cannot”; or (6) ”I do not feel severe pain”. The participants chose one of them that was deemed most applicable.

#### 2.3.2. Work-Related Factors

The participants were asked about their employment status (regular or casual, that is, a temporary contract and part-time job), occupation (teacher, cook/nutritionist, and others), and work schedule (regular or irregular).

#### 2.3.3. Demographic Characteristics

The participants were asked about their sex, age, height, and weight. Their height and weight were used to calculate their body mass index (BMI).

### 2.4. Statistical Analyses

Baseline information about their demographic characteristics and work-related factors was described for the 446 baseline subjects. We also described the details of the LBP of those with LBP at baseline. We calculated LBP prevalence at 1, 3, and 5 years among those with LBP at baseline. We also estimated the prevalence of LBP, assuming that LBP persisted among the dropouts at the same frequency as the participants of the corresponding follow-up surveys. Multiple logistic regression analysis was used to examine the association between the NRS (5 or greater) and RDQ (4 or greater) scores at baseline and the presence of LBP at 1, 3, and 5 years. The odds ratios were calculated and adjusted for sex, age, BMI, employment status, occupation, and work schedule. Those who did not participate in the surveys performed at 1, 3, and 5 years were excluded from the calculation of odds ratios. We conducted sensitivity analyses where only the female subjects were analyzed. For assessing potential selection biases, we compared the baseline information between the participants and dropouts for each follow-up survey.

## 3. Results

Table 1 shows the baseline information about demographic characteristics and work-related factors of the subjects (n = 446). Most participants were female (89.7%), regular employees (87.4%), teachers (85.7%), and working on an irregular schedule (81.8%). The ages of the participants ranged from 20 to 67, with the median being 31. Being overweight or obese, as defined with a BMI of 25 or greater, was reported by 9.2% of the subjects.

Figure 1 shows a flow of the follow-up survey participation. LBP was reported by 270 at baseline (60.5%, 95% confidence interval (95% CI): 56.0–65.1%). Of them, LBP was reported by 176 (65.2%), 139 (51.5%) and 109 (40.4%) at 1, 3, and 5 years, respectively. The numbers of dropouts were 62 (23.0%), 101 (37.4%) and 137 (50.7%) at 1, 3, and 5 years, respectively. Given that LBP persisted among the dropouts at the same frequency as the participants, the prevalence of persistent LBP was estimated to be 84.6% (80.3–88.9%), 82.2% (77.7–86.8%), and 82.0% (77.4–86.5%) at 1, 3, and 5 years, respectively. When only the female subjects were analyzed, the estimated prevalence of persistent LBP changed very little: 84.8% (80.2–89.3%), 81.6% (76.7–86.5%), and 84.3% (79.8–88.9%) at 1, 3, and 5 years (Appendix A).

Of the subjects with LBP at baseline (n = 270), 18.1% complained of leg pain (Table 2). The prevalence percentages of having an NRS of 5 or greater and an RDQ score of 4 or greater were 31.5% and 15.6%, respectively. Regarding the impact of LBP on work, 74.4% reported that “I feel slight pain occasionally”.

Table 3 and Table 4 present the odds ratios of the baseline NRS and RDQ scores for persistent LBP at 1, 3, and 5 years. Those with an NRS score of 5 or greater at the baseline were more likely to suffer from persistent LBP at 1 and 3 years, but this was not true for persistent LBP at 5 years. RDQ scores of 4 or greater at the baseline did not predict the presence of LBP at 1, 3, and 5 years. Sex, age, BMI, employment status, occupation, and work schedule at the baseline were not related to LBP at 1, 3, and 5 years (Appendix A). The odds ratios of the baseline NRS and RDQ scores for persistent LBP at 1, 3, and 5 years changed very little even when only the female subjects were analyzed (Appendix A).

Among those with LBP at baseline, there were no differences in the baseline characteristics between the participants and dropouts in the follow-up survey at 1 year. There were differences in age and RDQ scores between the participants and dropouts in the follow-up survey at 3 years. The participants were older (mean (SD): 37.1 (11.8) vs. 33.3 (12.3) years, *p* = 0.014) and showed a higher prevalence of an RDQ score of 4 or greater (19.0% vs. 10.0%, *p* = 0.049) at baseline than the dropouts. There were no differences in the other baseline characteristics. In the follow-up survey at 5 years, the participants were older than the dropouts (37.3 (11.1) vs. 34.0 (12.8) years, *p* = 0.025), while there were no differences in the other baseline characteristics.

## 4. Discussion

A previous systematic review found that LBP persisted for one year in 65% of patients with nonspecific LBP who were cared for by a primary healthcare physician [16]. In our study, we estimated a higher prevalence of persistent LBP, approximately 80%. This difference might come from our definition of LBP, which does not consider the degree of pain and limitation of daily activities due to LBP. The subjects were all working individuals in our study. The median NRS and RDQ scores were 3 and 1 in this study, respectively. They were lower than those of patients [24,26] and community residents [27] with LBP in Japan. Therefore, those who were diagnosed with LBP in our study tended to be mild cases. On the other hand, approximately 35% of our subjects with LBP at baseline had an NRS score of 5 or higher, and approximately 15% of them had an RDQ score of 4 or higher. This indicates that, although the NRS and RDQ scores were low on average, our subjects had a certain prevalence of clinically significant pain and serious disability in daily living due to LBP. A previous review reported that the prevalence of sciatica ranged from 1.2% to 43% [28]. The prevalence of leg pain in our study, approximately 20%, was within the range.

Here, we discuss the selection biases that could have occurred at the baseline and the potential effects on the present findings. We employed convenient sampling for the baseline subjects. Women accounted for as much as 89.7% of the sample. This share was lower than the average in Japan, 97%, and in the OECD countries, 96% [5]. In general, women show a higher prevalence of LBP than men [1]. A recent systematic review pointed out that female sex is a risk factor for the chronicity of LBP [19]. However, the prevalence of persistent LBP changed very little when we only analyzed the female subjects.

We also need to discuss the selection biases that could have occurred through the follow-up surveys and the potential effects on the present findings. An important fact is that 23.0%, 37.4%, and 50.7% of the initial participants dropped out at 1, 3, and 5 years, respectively. In order not to underestimate LBP prevalence, we estimated its prevalence at 1, 3, and 5 years, assuming that LBP similarly persisted among the dropouts at the same frequency as the participants. To justify this idea, we compared the baseline characteristics between the participants and dropouts. We found that there was no significant difference between the participants and dropouts, except for age and the RDQ score. The dropouts were only 3–4 years younger than the participants. The average age of both the participants and dropouts was in their mid-30s. This indicates that there was not a substantial selection bias in terms of age. In addition, in the follow-up survey conducted at 3 years, the participants presented a doubled prevalence of RDQ scores of 4 or greater, indicating a disability due to LBP, compared to the dropouts at baseline (19.0% vs. 10.0%). However, the RDQ scores were not associated with persistent LBP in our study. We found the differences in baseline characteristics between participants and dropouts to be minor, thus justifying our way of estimating LBP prevalence at 1, 3, and 5 years.

In this study, severe LBP pain, indicated as an NRS score of 5 or greater, at baseline predicted the presence of LBP at 1 and 3 years. This is concordant with the existing systematic review reporting that high pain intensity seems to increase the risk of chronic LBP [19]. Another systematic review pointed out that pain intensity potentially predicts the return to work of workers who leave work due to chronic LBP [17]. Our findings suggest that even nursery school workers with severe LBP should be monitored to prevent persistent LBP. On the other hand, our findings do not show that NRS scores predicted persistent LBP at 5 years. More studies are necessary to test whether NRS scores have a long-term predictive ability for LBP since there have been few relevant studies to our knowledge.

We failed to present in this study that disability in daily life due to LBP, indicated as an RDQ score of 4 or higher, at baseline predicted the persistence of LBP. We could not find existing evidence showing that RDQ scores predict persistent LBP. We only found previous studies reporting that RDQ scores are not associated with the pain intensity of LBP [29] or health-related quality of life [30]. It should be studied whether disability in daily life due to LBP is related to persistent LBP.

This study has some limitations besides the potential selection biases due to the sample collection and dropouts already discussed. A limitation of our LBP diagnosis was that we used a self-administered questionnaire, which could not distinguish between acute and chronic LBP or specify the cause of the LBP. Some LBP cases may not be caused by orthopedic and musculoskeletal conditions. One of the limitations regarding the subjects was that we did not examine the reasons for dropout. The dropouts could have quit working for reasons other than LBP. They might have continued to work but did not participate in the follow-up survey for personal reasons. They may have been temporarily away from the workplace for maternity or paternity leave, which is common among young female workers in Japan. Another limitation was employing convenience sampling. It cannot be completely guaranteed that the subjects in our study are representative samples of Japanese nursery school workers.

## 5. Conclusions

We conducted a five-year panel study on nursery school workers to describe the natural course of LBP and to examine whether pain intensity and disability in daily life due to LBP predicted persistent LBP. We discovered that once LBP occurred, it tended to persist for a long time. This highlights the importance of the primary prevention of LBP. Further research is needed to identify the preventive factors for primary LBP among nursery school workers. We also found that pain intensity could be a useful predictor of persistent LBP in this group. Measuring the NRS score is simple. We recommend it to policymakers and healthcare workers associated with nursery schools for the tertiary prevention of LBP.

## Figures and Tables

**Figure 1 healthcare-12-00128-f001:**
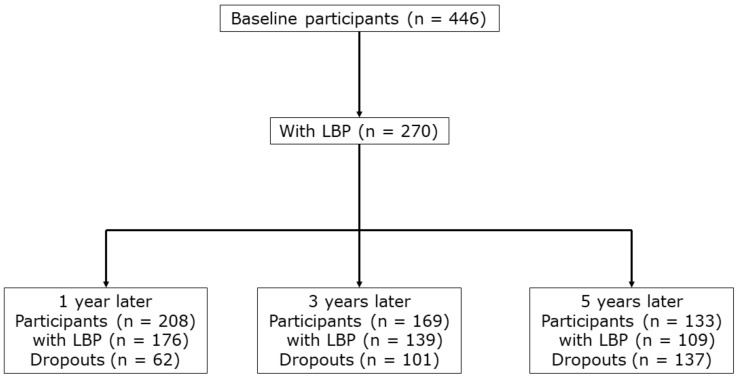
Flow of the baseline and follow-up surveys.

**Table 1 healthcare-12-00128-t001:** Demographic characteristics and work-related factors at baseline (n = 446).

Sex: female	400 (89.7%)
Age (median, range)	31.0 (20–67)
Body mass index (BMI) ≥ 25	41 (9.2%)
Work-related factors	
Employment status: regular	390 (87.4%)
Occupation	
Teacher	382 (85.7%)
Cook/Nutritionist	46 (10.3%)
Others	17 (3.8%)
Work schedule: irregular	365 (81.8%)
With low back pain	270 (60.5%)

Figures are presented as proportions (%), except for age. The numbers of missing responses were 5 for age, 15 for BMI, 2 for employment status, 1 for occupation, 3 for work schedule, and 2 for presence/absence of low back pain.

**Table 2 healthcare-12-00128-t002:** Details of low back pain at baseline (n = 270).

With leg pain	49 (18.1%)
Numerical rating scale score	3 (0–9)
5 or greater	85 (31.5%)
Roland–Morris Disability Questionnaire score	1 (0–19)
4 or greater	42 (15.6%)
Impact on work	
I cannot work without taking occasional days off.	0 (0%)
I cannot work without taking a break sometimes.	6 (2.2%)
It hurts a lot, but I do not need to take a break.	45 (16.7%)
I feel slight pain occasionally.	201 (74.4%)
I would like a break or day off, but I cannot.	5 (1.9%)
I do not have severe pain.	11 (4.1%)

Figures are presented as proportions (%) or medians (ranges).

**Table 3 healthcare-12-00128-t003:** Relationship between the numerical rating scale (NRS) scores at baseline and low back pain (LBP) at 1, 3, and 5 years.

NRS Score at Baseline	N (%) of Those with LBP	Adjusted Odds Ratio (95% Confidence Interval) ^(1)^
1 year later		
4 or less (n = 142)	115 (81.0)	1
5 or greater (n = 61)	57 (93.4)	4.01 (1.27–12.6) *
3 years later		
4 or less (n = 111)	84 (75.7)	1
5 or greater (n = 53)	51 (96.2)	8.51 (1.87–38.7) **
5 years later		
4 or less (n = 90)	69 (76.7)	1
5 or greater (n = 40)	37 (92.5)	3.49 (0.95–12.8)

*: *p* < 0.05; **: *p* < 0.01. The NRS scores were unavailable from 5, 5, and 3 participants for the results on LBP at 1, 3, and 5 years, respectively. ^(1)^ Sex, age, body mass index, employment status, occupation, and work schedule at the baseline were adjusted.

**Table 4 healthcare-12-00128-t004:** Relationship between the Roland–Morris Disability Questionnaire (RDQ) scores at baseline and low back pain (LBP) at 1, 3, and 5 years.

RDQ Score at Baseline	N (%) of Those with LBP	Adjusted Odds Ratio (95% Confidence Interval) ^(1)^
1 year later		
3 or less (n = 172)	143 (83.1%)	1
4 or greater (n = 35)	32 (91.4%)	1.92 (0.52–7.03)
3 years later		
3 or less (n = 136)	108 (79.4%)	1
4 or greater (n = 32)	30 (93.8%)	4.64 (0.98–22.0)
5 years later		
3 or less (n = 108)	85 (78.7%)	1
4 or greater (n = 24)	23 (95.8%)	7.17 (0.84–61.4)

The RDQ scores were unavailable from 1 participant for the present results. ^(1)^ Sex, age, body mass index, employment status, occupation, and work schedule at the baseline were adjusted.

## Data Availability

The data used in this study are available from the corresponding author upon reasonable request.

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
