# Peer review of "Persistence of Low Back Pain and Predictive Ability of Pain Intensity and Disability in Daily Life among Nursery School Workers in Japan: A Five-Year Panel Study"

_healthcare, 2024, doi:10.3390/healthcare12020128_

Round 1

Reviewer 1 Report

Comments and Suggestions for Authors

This study examines persistence of chronic low back pain among nursery school workers. The design is prospective with a 5-year follow up.

Introduction:

The introduction covers persistence of pain in the literature. However, determinants are not adequately covered. Please refer to the relevant literature about determinants of chronicity and persistence.

Methods:

Why was convenient sample used? Please describe in details the recruitment process. How were workers willing to participate spotted and recruited?

I would not assume that any person with back pain extending to the buttock, thigh or legs has sciatica. I think this requires confirmation by physical exam. Therefore, I would recommend you use back pain associated with leg pain instead of "sciatica", or back pain associated with referred pain.

Discussion:

Your sample is predominantly female. This warrants a discussion of the gender aspect of chronic low back pain. It is also a limitation to the generalization of your results.

Author Response

Reviewer 1

We appreciate your feedback on our manuscript. We have highlighted the revisions in yellow in the revised manuscript.

(Comment)

This study examines persistence of chronic low back pain among nursery school workers. The design is prospective with a 5-year follow up.

Introduction:

The introduction covers persistence of pain in the literature. However, determinants are not adequately covered. Please refer to the relevant literature about determinants of chronicity and persistence.

(Reply)

We added a sentence to emphasize the potential risk factors for chronicity of low back pain (LBP).

Lines 70 – 72

A recent systematic review found that female sex, high pain intensity, high body weight, carrying heavy loads at work, and difficult working positions were potential risk factors for chronicity of LBP [19].

(Comment)

Methods:

Why was convenient sample used? Please describe in details the recruitment process. How were workers willing to participate spotted and recruited?

(Reply)

We detailed the explanation of why we conducted a convenience sampling. This is because we had no way to access a population of nursery school workers large enough to conduct random sampling. We ask for an association of private nursery school facilities. We added a description of how to let the workers know about our survey and how we spotted and recruited those who were willing to participate.

Lines 89 – 97

Convenience sampling was performed since we had no way to access a population of nursery school workers large enough to conduct random sampling. We asked an association consisting of 36 private nursery school facilities in Nagoya, Japan, and its suburbs to recruit the participants. The inclusion criterion was being employed in nursery schools when the baseline survey was conducted. The exclusion criterion was leaving work for health-related or any other reasons when the baseline and follow-up surveys were conducted. The managers of the nursery schools introduced the survey to the workers orally and by written notice. The workers who were willing to participate in the survey gave us their consent.

(Comment)

I would not assume that any person with back pain extending to the buttock, thigh or legs has sciatica. I think this requires confirmation by physical exam. Therefore, I would recommend you use back pain associated with leg pain instead of "sciatica", or back pain associated with referred pain.

(Reply)

We replaced “sciatica” with “leg pain.” (Lines 118, 185, and 242 and Table 2)

(Comment)

Discussion:

Your sample is predominantly female. This warrants a discussion of the gender aspect of chronic low back pain. It is also a limitation to the generalization of your results.

(Reply)

We revised our discussion on the selection biases that could have occurred at the baseline and the potential effects on the present findings in terms of female sex.

Lines 244 – 250

Here, we discuss the selection biases that could have occurred at the baseline and the potential effects on the present findings. We employed convenient sampling for the baseline subjects. Women accounted for as much as 89.7% of the sample. This share was lower than the average in Japan, 97%, and in the OECD countries, 96% [5]. In general, women show a higher prevalence of LBP than men [1]. A recent systematic review pointed out that female sex is a risk factor for the chronicity of LBP [19]. However, the prevalence of persistent LBP changed very little when we only analyzed the female subjects.

Reviewer 2 Report

Comments and Suggestions for Authors

Dear authors,

The main initial observation I would like to make is about the design of the study. In my understanding, this is not a prospective cohort study, but a panel study. I therefore believe that a review of the study's methodological basis is necessary.

Another point is the multiple regression analysis carried out. I was unable to observe the final model constructed. Was no explanatory variable (sex, age, height, or weight) significant?

Author Response

Reviewer 2

We appreciate your feedback on our manuscript. We have highlighted the revisions in yellow in the revised manuscript.

(Comment)

Dear authors,

The main initial observation I would like to make is about the design of the study. In my understanding, this is not a prospective cohort study, but a panel study. I therefore believe that a review of the study's methodological basis is necessary.

(Reply)

We replaced “prospective cohort study” with “panel study” (Abstract: line 18, Keywords, and Text; lines 74, 83, and 296). A general definition of a panel study is that they collect repeated measures from the same sample at different points in time. [Heather Laurie. Panel Studies. In Oxford Bibliographies. 2013 DOI: 10.1093/obo/9780199756384-0108. Accessed on Dec. 20, 2023.]

We modified the title to “Persistence of low back pain and predictive ability of pain intensity and disability of daily life among nursery school workers in Japan: A five-year longitudinal study.”

(Comment)

Another point is the multiple regression analysis carried out. I was unable to observe the final model constructed. Was no explanatory variable (sex, age, height, or weight) significant?

(Reply)

We added Supplementary Tables S1 and S2 to show the complete results of the multiple regression analyses. Sex, age, body mass index, employment status, occupation, and work schedule at the baseline were not related to LBP at 1, 3, and 5 years later (lines 205 – 207).

Supplementary Table S1. Adjusted odds ratios of baseline numerical rating scale (NRS) scores and other characteristics for low back pain (LBP) at 1, 3, and 5 years later: logistic regression analyses.

Supplementary Table S2. Adjusted odds ratios of baseline Roland–Morris Disability Questionnaire (RDQ) scores and other characteristics for low back pain (LBP) at 1, 3, and 5 years later: logistic regression analyses.

Reviewer 3 Report

Comments and Suggestions for Authors

This 5-year prospective cohort study aimed to estimate the prevalence rate of persistent low back pain (LBP) among a convenience sample of 446 nursery school workers in Japan. Additionally, the study aimed to identify the predictive ability of pain intensity and pain-related daily life disability for the persistence of LBP in this specific patient population. The research addresses a clinically relevant and important issue. However, there are several concerns that should be addressed before deeming the paper suitable for publication.

Introduction

1.     The introduction starts by stating that low back pain is a common injury, but it would be beneficial to provide a brief explanation of why low back pain is a significant concern. Including information about the impact of low back pain on individuals' daily lives, work productivity, and healthcare costs would strengthen the rationale for studying this issue among nursery school workers specifically.

2.     The authors present several prevalence statistics from cross-sectional studies in different countries. While these statistics help to highlight the high prevalence of low back pain in nursery school workers, it would be valuable to mention the sample sizes and demographic characteristics of the study populations in these studies, which would allow readers to better understand the generalizability of the findings and potential variations across different populations.

3.     It has been mentioned that few longitudinal studies have been conducted on nursery school workers, but the gap in knowledge that the current study aims to address was not explicitly highlighted. Clearly stating that the study intends to fill this gap by providing insights into the natural course of low back pain and identifying predictors of persistent pain in nursery school workers would help enhance the significance of the research.

4.     While this study is a prospective cohort analysis that aims to describe the persistent lower back pain among nursery school workers and examine the predictive ability of pain intensity and disability, it would be helpful to clearly state the specific research objectives. To provide a more focused direction to the readers, the research questions, or hypotheses that the study aims to answer should be clearly articulated.

5.     The introduction lacks a concluding paragraph discussing the significance and potential implications of the study findings. It would be valuable to briefly mention the potential impact of the study on developing interventions, policies, or guidelines to improve the occupational health and well-being of nursery school workers.

Methods

1.     Understanding the method of participant recruitment is crucial for assessing the generalizability and potential biases of the study findings. Could you please provide more information on how the participants were approached and recruited for this study?

2.     A clearer definition of low back pain should be provided. Also, authors may need to elaborate more on the inclusion and exclusion criteria to help the readers understand to whom these results would be applicable.

3.     Employment status can be an important factor that may influence the occurrence and persistence of low back pain, and exploring this relationship could provide valuable insights into occupational risk factors. Did this research project include an analysis of the potential association between employment status and the outcomes of interest?

Results and discussion

1.     A notable observation from the study is that out of the total 446 respondents, 400 were identified as female. Considering the significant gender disparity within the sample, it raises questions about the potential influence of gender on the study results. In light of this, it would be intriguing to ascertain whether a gender-based analysis was conducted to explore the prevalence of low back pain. Including a synopsis of the gender-specific findings in the results section would provide valuable insights into any disparities or associations between gender and the prevalence of low back pain in the studied population.

2.     Longitudinal studies commonly employ follow-up assessments to monitor changes and trends over time. Notably, in this study, a considerable number of dropouts were observed at each respective follow-up interval. Specifically, there were 62 dropouts at the 1-year follow-up, 101 dropouts at the 3-year follow-up, and 137 dropouts at the 5-year follow-up, indicating relatively high dropout rates. Given these circumstances, it is pertinent to consider the potential impact of these dropouts on the reliability and validity of the results presented in this study.

3.     The discussion provides a thorough analysis of the study findings, comparing them to existing literature and highlighting areas that require further investigation. The limitations of the study are well-addressed, and future research directions are suggested. However, it would be beneficial to include a discussion on the clinical implications of the study findings and how they can contribute to the management and prevention of LBP among nursery school workers.

Author Response

Reviewer 3

We appreciate your feedback on our manuscript. We have highlighted the revisions in yellow in the revised manuscript.

(Comment)

This 5-year prospective cohort study aimed to estimate the prevalence rate of persistent low back pain (LBP) among a convenience sample of 446 nursery school workers in Japan. Additionally, the study aimed to identify the predictive ability of pain intensity and pain-related daily life disability for the persistence of LBP in this specific patient population. The research addresses a clinically relevant and important issue. However, there are several concerns that should be addressed before deeming the paper suitable for publication.

(Reply)

I appreciate your favorable evaluation. We have revised our manuscript, following the reviewers’ thoughtful and helpful comments.

(Comment)

Introduction

  1. The introduction starts by stating that low back pain is a common injury, but it would be beneficial to provide a brief explanation of why low back pain is a significant concern. Including information about the impact of low back pain on individuals' daily lives, work productivity, and healthcare costs would strengthen the rationale for studying this issue among nursery school workers specifically.

(Reply)

We modified the introduction and added some references. We detailed the years lived with disability (YLDs) due to low back pain (LBP) to give information about the impact of LBP on individuals’ daily lives. We mentioned sickness absences and return-to-work management for the work productivity of workers with LBP. We introduced systematic reviews on the associations between LBP and increased healthcare costs.

Lines 32 – 39

In 2020, it (LBP) affected 619 million people worldwide [1]. The global age-standardized rate of years lived with disability (YLDs) due to LBP was 832 per 100,000 [1]. Occupational ergo-nomic factors are the leading factor for YLDs, accounting for them by as much as 22% [1]. Workers with LBP are more likely to take absences due to sickness and require re-turn-to-work management [2]. Systematic reviews showed that LBP is associated with increased healthcare costs not only in high-income countries but also in low- and mid-dle-income countries [3, 4].

Newly added references

  1. Schaafsma, F.G.; Anema, J.R.; van der Beek, A.J. Back pain: Prevention and management in the workplace. Best. Pract. Res. Clin. Rheumatol. 2015, 29, 483 – 494.
  2. Dagenais, S.; Caro, J.; Haldeman, S. A systematic review of low back pain cost of illness studies in the United States and internationally. Spine J. 2008, 8, 8 – 20.
  3. Fatoye, F.; Gebrye, T.; Mbada, C.E.; Useh, U. Clinical and economic burden of low back pain in low- and middle-income countries: a systematic review. BMJ Open 2023, 13, e064119.

(Comment)

  1. The authors present several prevalence statistics from cross-sectional studies in different countries. While these statistics help to highlight the high prevalence of low back pain in nursery school workers, it would be valuable to mention the sample sizes and demographic characteristics of the study populations in these studies, which would allow readers to better understand the generalizability of the findings and potential variations across different populations.

(Reply)

We added the information on the sample size, valid response rate, and the proportion of women for each reference.

Lines 52 – 62).

In Japan, Tsuboi et al. [7] and Kudo and Sasaki [8] reported point prevalence rates of 43.0% (n = 142) and 71.9% (n = 57; valid response rate: 63.2%; female: 100%), respectively. Yamamoto-Kataoka et al. reported one-year and lifetime prevalence rates of 41.8% and 83.7%, respectively (n = 154; valid response rate: 38%; female: 93.5%) [9]. Isono et al. revealed that the one-month and lifetime prevalence rates were 62.0% and 85.5%, respectively (n = 333; valid response rate: 56.9%; female: 100%) [10]. In an Italian study, 70.6% of nursery school workers with musculoskeletal disorders experienced pain in their lower back (n = 677; valid response rate: 76.6%; female: 100%) [11]. In Jordan, Alghwiri et al. found that the prevalence of LBP was 46% in women and 36% in men, with 55% of women and 49% of men reporting that LBP interfered with their work (n = 439; valid response rate: 88%; female: 77%) [12].

(Comment)

  1. It has been mentioned that few longitudinal studies have been conducted on nursery school workers, but the gap in knowledge that the current study aims to address was not explicitly highlighted. Clearly stating that the study intends to fill this gap by providing insights into the natural course of low back pain and identifying predictors of persistent pain in nursery school workers would help enhance the significance of the research.

(Reply)

In our understanding, the gap in knowledge means that a certain number of cross-sectional studies have been conducted while few longitudinal studies have been conducted and the natural course and the determinants for the chronicity of LBP consequently remain unknown. We modified the introduction to emphasize the gap.

Lines 62 – 73

Evidence is scarce on the natural course of LBP among nursery school workers since few longitudinal studies have been conducted for them to date. In general, pain in chronic LBP decreased by half in the first 6 weeks; after that, the decrease was gradual and pain persisted [15]. LBP persisted one year after onset in 65% of patients with non-specific LBP who were cared for by primary healthcare physicians [16]. These findings were not confirmed for nursery school workers.

Little is also known about the determinants of prolonged LBP in nursery school workers. Pain intensity and disability in daily life due to LBP are thought to affect the prognosis of LBP [17-19]. A recent systematic review found that female sex, high pain intensity, high body weight, carrying heavy loads at work, and difficult working positions were potential risk factors for chronicity of LBP [19]. However, there is little evidence to confirm whether this is true for nursery school workers.

(Comment)

  1. While this study is a prospective cohort analysis that aims to describe the persistent lower back pain among nursery school workers and examine the predictive ability of pain intensity and disability, it would be helpful to clearly state the specific research objectives. To provide a more focused direction to the readers, the research questions, or hypotheses that the study aims to answer should be clearly articulated.

(Reply)

We added a sentence to the last paragraph to emphasize our research question and purpose.

Lines 76 – 80

Our purpose was to elucidate how much LBP persists among nursery school workers and whether nursery school workers with severe pain intensity and disability in daily life due to LBP suffer from persistent LBP.

(Comment)

  1. The introduction lacks a concluding paragraph discussing the significance and potential implications of the study findings. It would be valuable to briefly mention the potential impact of the study on developing interventions, policies, or guidelines to improve the occupational health and well-being of nursery school workers.

(Reply)

We added a sentence to the last paragraph to highlight the applicability of our findings for policymakers and healthcare workers.

Lines 81 - 83

Policymakers and healthcare workers could utilize these findings to create strategies and implement preventative measures for LBP.

(Comment)

Methods

  1. Understanding the method of participant recruitment is crucial for assessing the generalizability and potential biases of the study findings. Could you please provide more information on how the participants were approached and recruited for this study?

(Reply)

We revised the explanation to detail how the participants were approached and recruited for this study.

Lines 89 – 92

Convenience sampling was performed since we had no way to access a population of nursery school workers large enough to conduct random sampling. We asked an association consisting of 36 private nursery school facilities in Nagoya, Japan, and its suburbs to recruit the participants.

Lines 95 - 97

The managers of the nursery schools introduced the survey to the workers orally and by written notice. The workers who were willing to participate in the survey gave us their consent.

(Comment)

  1. A clearer definition of low back pain should be provided. Also, authors may need to elaborate more on the inclusion and exclusion criteria to help the readers understand to whom these results would be applicable.

(Reply)

We made revisions to make the inclusion and exclusion criteria (lines 92 – 95) and the definition of LBP (lines 114 – 122) clearer.

Lines 92 – 95

The inclusion criterion was being employed in nursery schools when the baseline survey was conducted. The exclusion criterion was leaving work for health-related or any other reasons when the baseline and follow-up surveys were conducted.

Lines 111 – 119

We determined whether the participants had LBP based on their self-report to the statement “Specify where you are currently feeling LBP,” which included the following five sub-questions: (1) feeling pain in the lower back only, (2) feeling pain in the lower and upper back, (3) feeling pain that extends to the buttock and thighs, (4) feeling pain and numbness that extends to legs and feet, and (5) feeling pain in the shoulders, neck, and/or arms as well as the lower back. The participants gave “yes” or “no” to each of the sub-questions. Those who answered “yes” to any sub-question were defined as suffering from LBP. The subjects were regarded as having LBP combining leg pain when they answered affirmatively to either or both of sub-questions (3) and (4).

(Comment)

  1. Employment status can be an important factor that may influence the occurrence and persistence of low back pain, and exploring this relationship could provide valuable insights into occupational risk factors. Did this research project include an analysis of the potential association between employment status and the outcomes of interest?

(Reply)

The employment status (regular, casual), occupation (teacher, cook/nutritionist/other), and work schedule (regular, irregular) at the baseline were included as the independent factors in the multiple logistic regression models to examine the relationship between the baseline numerical rating scale (NRS) and Rolland-Morris Disability Questionnaire (RDQ)scores and LBP at 1, 3, and 5 years later. None of the employment status, occupation, or work schedule at the baseline was related to LBP at 1, 3, and 5 years later. To make it clear, we modified the materials and methods (lines 158 – 159) and results (lines 206 – 208) and added the supplementary tables S1 and S2.

Lines 154 – 155

The odds ratios were calculated and adjusted for sex, age, BMI, employment status, occupation, and work schedule.

Lines 205 – 207

Sex, age, body mass index, employment status, occupation, and work schedule at the baseline were not related to LBP at 1, 3, and 5 years later (Supplementary Tables S1 and S2).

Supplementary Table S1. Adjusted odds ratios of baseline numerical rating scale (NRS) scores and other characteristics for low back pain (LBP) at 1, 3, and 5 years later: logistic regression analyses.

Supplementary Table S2. Adjusted odds ratios of baseline Roland–Morris Disability Questionnaire (RDQ) scores and other characteristics for low back pain (LBP) at 1, 3, and 5 years later: logistic regression analyses.

(Comment)

Results and discussion

  1. A notable observation from the study is that out of the total 446 respondents, 400 were identified as female. Considering the significant gender disparity within the sample, it raises questions about the potential influence of gender on the study results. In light of this, it would be intriguing to ascertain whether a gender-based analysis was conducted to explore the prevalence of low back pain. Including a synopsis of the gender-specific findings in the results section would provide valuable insights into any disparities or associations between gender and the prevalence of low back pain in the studied population.

(Reply)

We conducted sensitivity analyses where only the female subjects were analyzed. The results changed little. We discussed the potential effect of the dominance of the female in our sample on the present findings. Finally, we made some revisions and added some description on this point and supplementary figure and tables.

Materials and methods, line 157

We conducted sensitivity analyses where only the female subjects were analyzed.

Results, lines 172 – 175

When only the female subjects were analyzed, the estimated prevalence of persistent LBP changed very little: 84.8% (80.2 – 89.3%), 81.6% (76.7 – 86.5%), and 84.3% (79.8 – 88.9%) at 1, 3, and 5 years later (Supplementary Figure S1).

Results, lines 207 – 209

The odds ratios of the baseline NRS and RDQ scores for persistent LBP at 1, 3, and 5 years later changed very little even when only the female subjects were analyzed (Supplementary Tables S3 and S4).

Discussion, lines 244 – 250

Here, we discuss the selection biases that could have occurred at the baseline and the potential effects on the present findings. We employed convenient sampling for the baseline subjects. Women accounted for as much as 89.7% of the sample. This share was lower than the average in Japan, 97%, and in the OECD countries, 96% [5]. In general, women show a higher prevalence of LBP than men [1]. A recent systematic review pointed out that female sex is a risk factor for the chronicity of LBP [19]. However, the prevalence of persistent LBP changed very little when we only analyzed the female subjects.

Supplementary Figure S1. Flow of the baseline and follow-up surveys: female subjects.

Supplementary Table S3. Relationship between the numerical rating scale (NRS) scores at baseline and low back pain (LBP) at 1, 3, and 5 years later among only female subjects.

Supplementary Table S4. Relationship between the Roland–Morris Disability Questionnaire (RDQ) scores at baseline and low back pain (LBP) at 1, 3, and 5 years later among only female subjects.

(Comment)

  1. Longitudinal studies commonly employ follow-up assessments to monitor changes and trends over time. Notably, in this study, a considerable number of dropouts were observed at each respective follow-up interval. Specifically, there were 62 dropouts at the 1-year follow-up, 101 dropouts at the 3-year follow-up, and 137 dropouts at the 5-year follow-up, indicating relatively high dropout rates. Given these circumstances, it is pertinent to consider the potential impact of these dropouts on the reliability and validity of the results presented in this study.

(Reply)

We modified our discussion on the selection biases that the dropouts could have occurred and the potential effects on the present findings to make our treatment more understandable and justified.

Lines 251 – 266

We also need to discuss the selection biases that could have occurred through the follow-up surveys and the potential effects on the present findings. An important fact is that 23.0%, 37.4%, and 50.7% of the initial participants dropped out 1, 3, and 5 years later, respectively. In order not to underestimate LBP prevalence, we estimated the prevalence of LBP at 1, 3, and 5 years later, assuming that LBP similarly persisted among the dropouts at the same frequency as the participants. To justify this idea, we compared the baseline characteristics between the participants and dropouts. We found that there was no significant difference between the participants and dropouts, except for age and the RDQ score. The dropouts were only 3-4 years younger than the participants. The average age of both the participants and dropouts was in their mid-30s. This indicates that there was not a substantial selection bias in terms of age. In addition, in the follow-up survey conducted 3 years later, the participants presented a doubled prevalence of RDQ scores of 4 or greater, indicating a disability due to LBP, compared to the dropouts at baseline (19.0% vs. 10.0%). However, the RDQ scores were not associated with persistent LBP in our study. We found the differences in baseline characteristics between participants and dropouts minor, justifying our way of estimating LBP prevalence at 1, 3, and 5 years later.

(Comment)

  1. The discussion provides a thorough analysis of the study findings, comparing them to existing literature and highlighting areas that require further investigation. The limitations of the study are well-addressed, and future research directions are suggested. However, it would be beneficial to include a discussion on the clinical implications of the study findings and how they can contribute to the management and prevention of LBP among nursery school workers.

(Reply)

We added the clinical implications and applications of our findings to the conclusions.

Lines 298 – 303

We discovered that once LBP occurred, it tended to persist for a long time. This highlights the importance of the primary prevention of LBP. Further research is needed to identify the preventive factors for primary LBP among nursery school workers. We also found that pain intensity could be a useful predictor of persistent LBP in this group. Measuring the NRS score is simple. We recommend it to policymakers and healthcare workers associated with nursery schools for the tertiary prevention of LBP.

Reviewer 4 Report

Comments and Suggestions for Authors

Overall, a generally well written and interesting paper. 

Line 30: suggest change from injury to condition. LBP isnt the injury, it is the condition that is caused by an injury. One of the main limitations is the introduction. It does not sufficiently support the study.

For example, Why study nursery school workers? What is it about the population that makes it important to look at this group? What s the evidence to suggest that LBP is a problem? How do LBP injuries occur in this population? - more evidence is needed in your introduction to justify your population.  Some further minor comments are also below. 

Methods: Was the survey a paper survey? How was it provided to the participants? How did they complete it? And return it to researchers? 

The results are well presented. The tables are easy to read. Did you consider doing any significance testing on the variables? For example, is there a significant relationship between BMI >25 and LBP? 

The first paragraph of your discussion is actually results. I would remove this and add it to the results section. Kepp your discussion around the high level applicability of your findings. 

Your conclusion is nicely aligned with your aim, although is it possible to draw more from your findings? What does this mean for nursery school workers? Also, do you plan to do further study? 

Comments on the Quality of English Language

The language is generally fine. Some very minor things (irregular workers could be "casual" workers). 

Author Response

Reviewer 4

We appreciate your feedback on our manuscript. We have highlighted the revisions in yellow in the revised manuscript.

(Comment)

Overall, a generally well written and interesting paper.

(Reply)

I appreciate your favorable evaluation. We have revised our manuscript, following the reviewers’ thoughtful and helpful comments.

(Comment)

Line 30: suggest change from injury to condition. LBP isnt the injury, it is the condition that is caused by an injury.

(Reply)

We modified the sentence.

Line 33

Low back pain (LBP) is a common health condition.

(Comment)

One of the main limitations is the introduction. It does not sufficiently support the study.For example, Why study nursery school workers? What is it about the population that makes it important to look at this group? What s the evidence to suggest that LBP is a problem? How do LBP injuries occur in this population? - more evidence is needed in your introduction to justify your population.  Some further minor comments are also below.

(Reply)

We added a paragraph and references to justify why we examined low back pain (LBP) of nursery school teachers.

Lines 40 – 50

The importance of early childhood education and care is widely acknowledged. Early childhood education and care contribute to children’s cognitive and emotional development, learning, and well-being [5]. Simultaneously, researchers have long warned about the health and safety risks that nursery school workers are exposed to in their work environment and the health concerns resulting from the risks, i.e., infectious diseases, musculoskeletal injuries, accidents, and occupational stress [6]. Nursery school workers are exposed to the following harmful ergonomic factors while working: heavy lifting, pushing and applying force, frequent bending and twisting, awkward standing and posture, sud-den load bearing, and repetitive work [6]. Since these are the supposed risk factors of LBP, LBP has attracted attention as a prevalent health concern among nursery school workers.

Newly added references

  1. OECD. Indicator B2. How do early childhood education systems differ around the world? In Education at a Glance 2023: OECD Indicators.; OECD Publishing: Paris, France, 2023, pp. 166 – 190.
  2. McGrath, B.J. Identifying health and safety risks for childcare workers. AAOHN J. 2007, 55, 321 – 325; quiz 326 – 327.

(Comment)

Methods: Was the survey a paper survey? How was it provided to the participants? How did they complete it? And return it to researchers?

(Reply)

Our survey was a paper survey. We cooperated with a printing company, Tokai Kyodo Printing Inc., for data collection and editing as we wrote in the Acknowledgments (lines 328 – 329). We add the relevant explanation in the Materials and Methods.

Lines 97 – 99

A bubble sheet that also contained the questions was sent to the participants by a printing company. They filled it in and sent it back to the printing company for data entry.

(Comment)

The results are well presented. The tables are easy to read. Did you consider doing any significance testing on the variables? For example, is there a significant relationship between BMI >25 and LBP?

(Reply)

We added Supplementary Tables S1 and S2 to show the complete results of the multiple logistic regression analyses. Sex, age, body mass index, employment status, occupation, and work schedule at the baseline were adjusted for the multiple logistic regression analyses and were not related to LBP at 1, 3, and 5 years later. To make it clearer, we made some modifications.

Lines 154 – 155, Materials and methods

The odds ratios were calculated and adjusted for sex, age, BMI, employment status, occupation, and work schedule.

Lines 205 – 207, Results

Sex, age, body mass index, employment status, occupation, and work schedule at the baseline were not related to LBP at 1, 3, and 5 years later (Supplementary Tables S1 and S2).

Supplementary Table S1. Adjusted odds ratios of baseline numerical rating scale (NRS) scores and other characteristics for low back pain (LBP) at 1, 3, and 5 years later: logistic regression analyses.

Supplementary Table S2. Adjusted odds ratios of baseline Roland–Morris Disability Questionnaire (RDQ) scores and other characteristics for low back pain (LBP) at 1, 3, and 5 years later: logistic regression analyses.

(Comment)

The first paragraph of your discussion is actually results. I would remove this and add it to the results section. Kepp your discussion around the high level applicability of your findings.

(Reply)

The paragraph appears repetitive to us, as you commented. We eventually removed it.

(Comment)

Your conclusion is nicely aligned with your aim, although is it possible to draw more from your findings? What does this mean for nursery school workers? Also, do you plan to do further study?

(Reply)

Following your comment, we presented the potential application of our findings.

Lines 298 – 303

We discovered that once LBP occurred, it tended to persist for a long time. This highlights the importance of the primary prevention of LBP. Further research is needed to identify the preventive factors for primary LBP among nursery school workers. We also found that pain intensity could be a useful predictor of persistent LBP in this group. Measuring the NRS score is simple. We recommend it to policymakers and healthcare workers associated with nursery schools for the tertiary prevention of LBP.

(Comment)

The language is generally fine. Some very minor things (irregular workers could be "casual" workers).

(Reply)

We asked the MDPI’s English language editing service to proofread our manuscript. Grammatical modifications are highlighted in yellow in the revised manuscript.

Round 2

Reviewer 2 Report

Comments and Suggestions for Authors

Dear authors,

Thank you for returning to the considerations made in the first version.

I only have one residual observation: I don't understand why the study was kept longitudinal in the title of the manuscript, although it is classified as a panel study. By the way, throughout the body text is treated as a panel.

Author Response

We modified the title to “Persistence of low back pain and predictive ability of pain intensity and disability of daily life among nursery school workers in Japan: A five-year panel study.”

We appreciate your feedback on our revised manuscript. We have highlighted the revisions in yellow in the title.

Reviewer 3 Report

Comments and Suggestions for Authors

The authors have adeptly tackled all previous concerns. I have no further remarks to add.

Author Response

We are delighted to receive your positive feedback regarding our revised manuscript.